# Photoelectric responsive ionic channel for sustainable energy harvesting

Qing Guo[1], Zhuozhi Lai[1], Xiuhui Zuo[1], Weipeng Xian[1], Shaochun Wu[1], Liping Zheng[2], Zhifeng Dai [2], Sai Wang[1] ✉ & Qi Sun [1] ✉

Access to sustainable energy is paramount in today's world, with a significant emphasis on solar and water-based energy sources. Herein, we develop photo-responsive ionic dye-sensitized covalent organic framework membranes. These innovative membranes are designed to significantly enhance selective ion transport by exploiting the intricate interplay between photons, electrons, and ions. The nanofluidic devices engineered in our study showcase exceptional cation conductivity. Additionally, they can adeptly convert light into electrical signals due to photoexcitation-triggered ion movement. Combining the effects of salinity gradients with photo-induced ion movement, the efficiency of these devices is notably amplified. Specifically, under a salinity differential of 0.5/0.01 M NaCl and light exposure, the device reaches a peak power density of 129 W m$^{-2}$, outperforming the current market standard by approximately 26-fold. Beyond introducing the idea of photoelectric activity in ionic membranes, our research highlights a potential pathway to cater to the escalating global energy needs.

The intricate processes found in nature provide a valuable blueprint for generating clean energy by seamlessly integrating multiple functions[1–3]. Replicating these natural mechanisms for renewable energy conversion through the synergistic fusion of various functionalities within artificial ion channels presents promising avenues for scientific exploration[4–14]. However, the creation of artificial nanochannels that harmoniously integrate diverse functions, enabling a synergistic response to external stimuli, poses a significant challenge[15]. It is well-established that both solar energy and salinity gradients play fundamental roles in driving ion transport, forming the basis for their integration and enabling the simultaneous harnessing of solar energy and osmotic power (Fig. 1a)[6–19]. Central to this endeavor is the pivotal scientific challenge of designing ionic membranes endowed with photo-responsive characteristics.

Recent advances in materials science have demonstrated significant potential in the development of nanofluidic devices capable of replicating the asymmetric ion transport and adaptive responsiveness observed in biological systems[20–28]. Nonetheless, relatively little attention has been dedicated to ionic membranes exhibiting photoelectric responsiveness[29]. To overcome the challenges associated with the time-consuming synthetic processes required for covalent grafting, we introduced a host-guest assembly strategy to create artificial nanochannels that emulate the functionality of their biological counterparts[30–34]. In pursuit of this goal, we opted for covalent organic frameworks (COFs) as our platform of choice, due to their tunability, enabling precise control over composition and nanopore architecture[35–45]. To further enhance the functionality of our system, we employed dyes as the preferred guest molecules. Dyes are renowned for their remarkable photoresponsive properties, easily excited upon absorbing light, facilitating photo-induced electron transfer, and generating a charge-separated state. Additionally, a substantial number of dyes are ionic, opening alternative avenues for creating ionic channels[46,47]. By merging the well-ordered nanoporosity of COFs with photo-responsive ionic dyes, we envisioned our system serving as an ideal platform for fundamental research on nanofluidic energy conversion.

[1]Zhejiang Provincial Key Laboratory of Advanced Chemical Engineering Manufacture Technology, College of Chemical and Biological Engineering, Zhejiang University, Hangzhou, China. [2]Key Laboratory of Surface & Interface Science of Polymer Materials of Zhejiang Province, Department of Chemistry, College of Science, Zhejiang Sci-Tech University, Hangzhou, China. ✉e-mail: wangsai@zju.edu.cn; sunqichs@zju.edu.cn

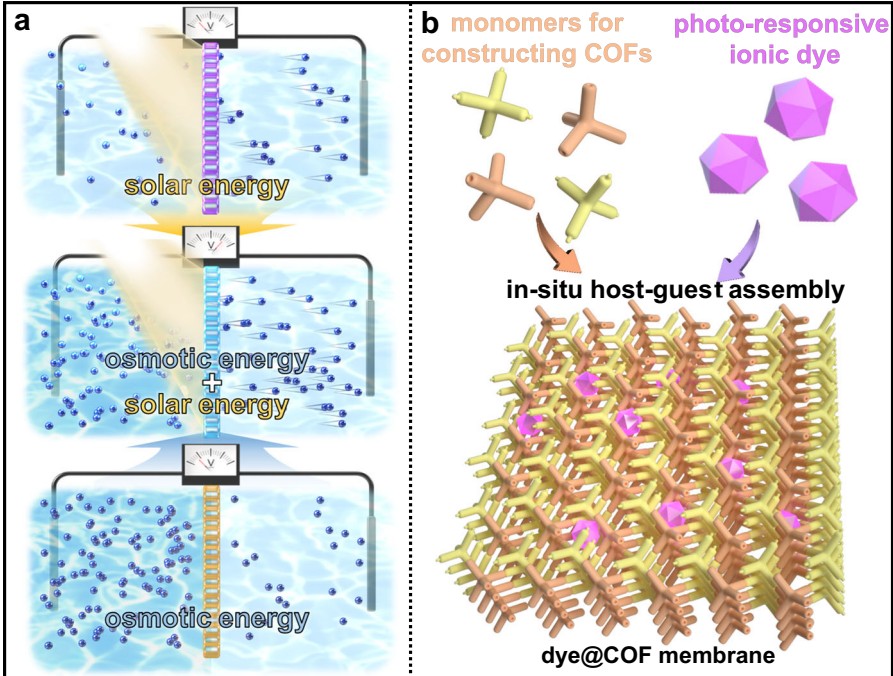

**Fig. 1 | Photoelectric responsive ionic channel for energy harvesting. a** Conceptual schematic of the consolidation of concentration gradient and light-driven ion transport for energy conversion, and **b** schematic of in-situ host-guest assembly for targeting ionic membranes with photo-responsive characteristics.

Capitalizing on these advantages, we successfully developed a series of photo-responsive ionic dye-sensitized COF membranes through an in-situ host-guest assembly process (Fig. 1b). This innovative approach harnesses the unique combination of photo-responsive ionic dye molecules and sub-nanosized COF pore channels to form a nanofluidic energy conversion device. Integrating ionic dye molecules into the COF structure conferred ion-screening capabilities upon the membrane. Consequently, the resulting composite membrane exhibited exceptional permselectivity across a wide range of electrolyte concentrations, underscoring its significant potential for osmotic energy harvesting. For a detailed explanation of the working principle, please refer to Supplementary Fig. 1. Furthermore, the photoelectric responsiveness of the dye molecules enabled these composite membranes to convert external light into electrical signals by facilitating photoexcitation-induced ion transmission. As a result, this bifunctional nanofluidic device allows the nanoconfined manipulation of ion diffusion, synergistically controlled by diverse driving forces. This synergy led to an impressive power density of up to 129 W m$^{-2}$, achieved under conditions simulating a concentration difference found at a river mouth (0.5/0.01 M NaCl) and exposure to light irradiation, thereby establishing a new benchmark (Supplementary Table 1). The versatility of our host-guest assembly strategy for creating multifunctional nanofluidic membranes underscores their significant potential for realizing emergent properties. Moreover, these membranes hold promise for future integration into devices designed for sustainable energy harvesting.

## Results

COF-301[48], a three-dimensional COF, synthesized through condensation of tetratopic amine and ditopic aldehyde—specifically, tetrakis(4- aminophenyl)methane (TAM) and 2,5-dihydroxyterephthaldehyde (DHA)—was chosen as the host material for encapsulating the dye due to its adherence to all necessary requirements. Notably, its angstrom-sized, interconnected pore channels thwart the leaching of nano-sized dye molecules, while its abundantly present hydroxyl groups potentially enhance ion transport via hydrogen bonding interactions between hydrated ions and channel walls

(Supplementary Fig. 2)[49]. The method for fabricating the COF-301 membrane was adapted from established literature, employing liquid−liquid interfacial polymerization[50]. During this process, a DHA mixture (in a mesitylene/ethyl acetate solvent) was gently introduced atop an aqueous solution of TAM in acetic acid. Verification of COF material formation was confirmed through observing a characteristic imine stretching vibration band at 1610 cm$^{-1}$ in the Fourier-transform infrared (FT-IR) spectra, indicative of amine-aldehyde condensation (Supplementary Fig. 3)[33]. N$_2$ sorption isotherms, analyzed at 77 K, disclosed a Brunauer−Emmett−Teller (BET) surface area of 564 m$^2$ g$^{-1}$ and a centered pore size of 7.5 Å (Supplementary Fig. 4). Powder X-ray diffraction (PXRD) experiments were conducted to assess the crystallinity of COF-301, revealing a set of peaks, wherein the positions and relative intensities of diffraction peaks concurred with the simulated structure (Supplementary Fig. 5).

To incorporate dye molecules into the membrane, initial studies focused on encapsulating hydroxynaphthol blue (HB), an anionic dye characterized by three sodium sulfonate groups and two hydroxyl groups within a single molecule, measuring approximately 1.5 × 1.0 nm in size (Supplementary Fig. 6). The loading of HB inside the COF could be finely controlled by adjusting the concentration of the dye in the initial reaction within a specific range. This led to the fabrication of HB$_x$@COF-301 membranes, with 'x' (mg mL$^{-1}$) referring to the concentration of HB used in the composite membrane synthesis. Four different loadings of HB were created, denoted as HB$_{1.9}$@COF-301, HB$_{3.8}$@COF-301, HB$_{5.7}$@COF-301, and HB$_{7.6}$@COF-301. The produced membranes underwent thorough washing with ethanol and deionized water to eliminate any unreacted monomers and loosely bound dye molecules. A visual examination of the membranes verified the successful encapsulation of HB molecules. Interestingly, while pristine COF-301 has an orange appearance, the HB$_x$@COF-301 membranes transformed into a deeper vermeil color, intensifying as the concentration of dye molecules increased (Supplementary Fig. 7). To comprehend the mechanism of dye encapsulation during membrane synthesis, the hypothesis proposed that this process was facilitated by electrostatic attraction between protonated TAM (TAM-acetic acid) molecules and negatively charged dye molecules. Ultraviolet−visible

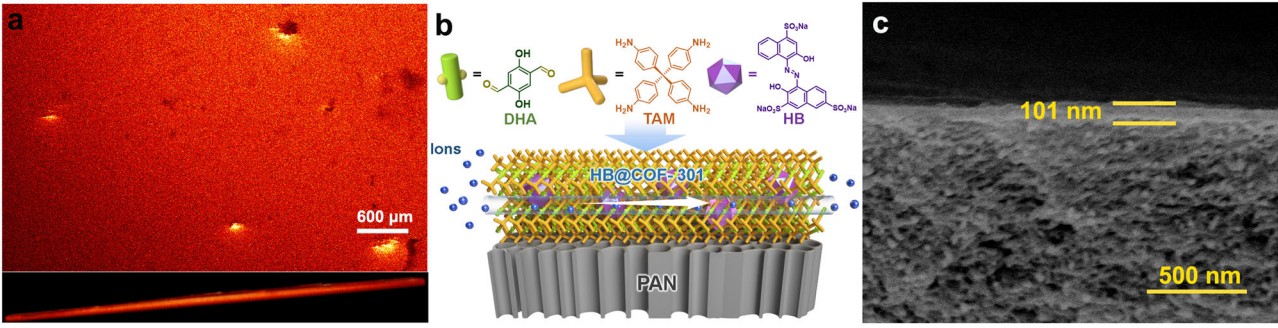

**Fig. 2 | Characterization of the composite membrane. a** Confocal microscopy images of the free-standing HB$_{5.7}$@COF-301 membrane (top and cross-sectional view). **b** Schematic of the structure of composite membranes. **c** Cross-sectional SEM image of HB$_{5.7}$@COF-301/PAN.

(UV–vis) spectroscopy was employed to validate this hypothesis. As expected, the maximum absorption peak of the HB aqueous solution exhibited a noticeable red-shift from 229 and 530 to 246 and 536 nm, respectively, following the addition of the TAM-acetic acid aqueous solution. This confirmed the interaction between these two compounds (Supplementary Fig. 8). The inclusion of HB molecules into the composites is further supported by FT-IR spectra, which revealed characteristic stretching vibrations at 1033 and 1105 cm$^{-1}$ for sulfonate groups.[51] Moreover, with an increase in the content of dye molecules, the relative intensity of the peaks associated with the sulfonate group also increased (Supplementary Fig. 9). For a visual representation of the spatial distribution of HB within the composites, confocal laser scanning microscopy (CLSM) was employed. For comparison, HB-on-COF-301 was synthesized by immersing a pre-synthesized COF-301 membrane in an HB aqueous solution. The images in Fig. 2a and Supplementary Figs. 10 and 11 highlighted clear disparities between the surface and cross-sectional CLSM images of HB$_{5.7}$@COF-301 and HB-on-COF-301. In HB$_{5.7}$@COF-301, HB molecules were evenly dispersed within the membrane, while only a faint fluorescence intensity was detected on the membrane surface for HB-on-COF-301. These findings provide substantial support for the proposed dye encapsulation mechanism. To quantitatively determine the HB content in the membranes, the COF hosts were digested with 1 M HCl, and the released dye concentrations were measured using UV–vis spectroscopy (Supplementary Fig. 12). As a result, the HB contents of HB$_{1.9}$@COF-301, HB$_{3.8}$@COF-301, HB$_{5.7}$@COF-301, and HB$_{7.6}$@COF-301 were estimated to be 0.181, 0.296, 0.324, and 0.337 mmol g$^{-1}$, respectively (Supplementary Table 2).

Subsequently, we initiated the growth of HB@COF-301 active layers on a polyacrylonitrile (PAN) ultrafiltration membrane to facilitate the assembly of nanofluidic devices. The selection of PAN was based on the rationale that it offers a highly porous structure, thus facilitating interfacial polymerization. Additionally, the cylindrical channel orientation of PAN is perpendicular to the ion transport direction after assembly into a nanofluidic device (Supplementary Figs. 13 and 14). This orientation implies that ions exclusively traverse the COF pore channels, simplifying the interpretation of experimental results (Fig. 2b). Similar to the observations made with free-standing HB$_x$@COF-301, the color of the HB$_x$@COF-301/PAN membranes deepened as the HB content increased (Supplementary Fig. 15). The integral area associated with the S species in the X-ray photoelectron spectra of HB$_x$@COF-301/PAN and the characteristic FT-IR peaks linked to sulfonate (O=S=O) increased with the concentration of the dye solutions used, suggesting the encapsulation of a higher quantity of HB molecules (Supplementary Figs. 16 and 17). The zeta potentials of the HB$_x$@COF-301/PAN membranes became more negative in comparison to the pristine PAN membrane. The zeta potential decreased from −19.1 to −48.9 mV as the HB content increased (Supplementary Table 3). Furthermore, water contact angle measurements

demonstrated that the composite membranes exhibited increased hydrophilicity, with contact angles decreasing from 82.4° to 62.5° with increasing HB content (Supplementary Fig. 18). Scanning electron microscopy (SEM) images of HB$_x$@COF-301/PAN unveiled the presence of seamless membrane surfaces with a thickness ranging from 90 to 101 nm, affixed to the PAN support (Fig. 2c and Supplementary Figs. 19–23).

To create a nanofluidic device based on COF composites, the membrane was initially cut into desirable sizes using a knife. In this study, we used rectangular strips measuring approximately 1.5 by 2.0 mm. These resulting strips were then assembled with a poly-dimethylsiloxane (PDMS) elastomer in a poly(methyl methacrylate) testing container. In the PDMS, two reservoirs with a volume of around 0.18 cm$^3$ were carved out on either side, exposing the two ends of the membrane to the electrolyte solution. A pair of Ag/AgCl electrodes was submerged into these reservoirs to establish the electrical circuit (Fig. 3a and Supplementary Fig. 24). The newly assembled devices were subsequently immersed in deionized water until the membrane reached full hydration, as indicated by the stabilization of the current. For PAN, no discernible current signal was detected, confirming the absence of any percolated leakage pathways along the direction of the ionic current. In contrast, after an overnight exposure to water, the composite membrane displayed a stable current signal, substantiating the viability of studying ion transport across dye-doped COF membranes (Supplementary Figs. 25 and 26).

To assess the charge selectivity of the nanofluidic device, we measured reversal potentials ($V_r$) on a representative device constructed using HB$_{5.7}$@COF-301/PAN when exposed to varying asymmetric electrolytes at concentrations of 0.1 M and 1 mM, including LiCl, NaCl, and KCl (Fig. 3b). According to the Goldman–Hodgkin–Katz equation[52], the device primarily conducts cations, demonstrating Li$^+$:Cl$^-$, Na$^+$:Cl$^-$, and K$^+$:Cl$^-$ selectivities of 125:1, 298:1, and 394:1, respectively, translating to a permselectivity exceeding 0.99. Consequently, the encapsulated negatively charged dye molecules effectively establish coion exclusion conditions. To elucidate the mechanism behind the ion selectivity of the dye-encapsulated membranes and emphasize the role of the dye molecules, we measured the ion conductance of the devices assembled using composite membranes with varying HB contents. We investigated trans-membrane ion transport behavior under identical KCl conditions by fitting the current–voltage ($I$–$V$) curves to the conductance model, with concentrations ranging from 0.01 mM to 3 M (Supplementary Fig. 27). Up to a KCl concentration of 1 M, the conductance maintains a plateau, irrespective of nominal ion concentrations, showcasing characteristic surface-charge-governed ion transport behavior (Supplementary Table 4). The plots illustrating conductivities against ion concentrations over HB$_x$@COF-301/PAN, as shown in Fig. 3c, can be rationalized as follows: the hosted HB molecules form a negative ion transport

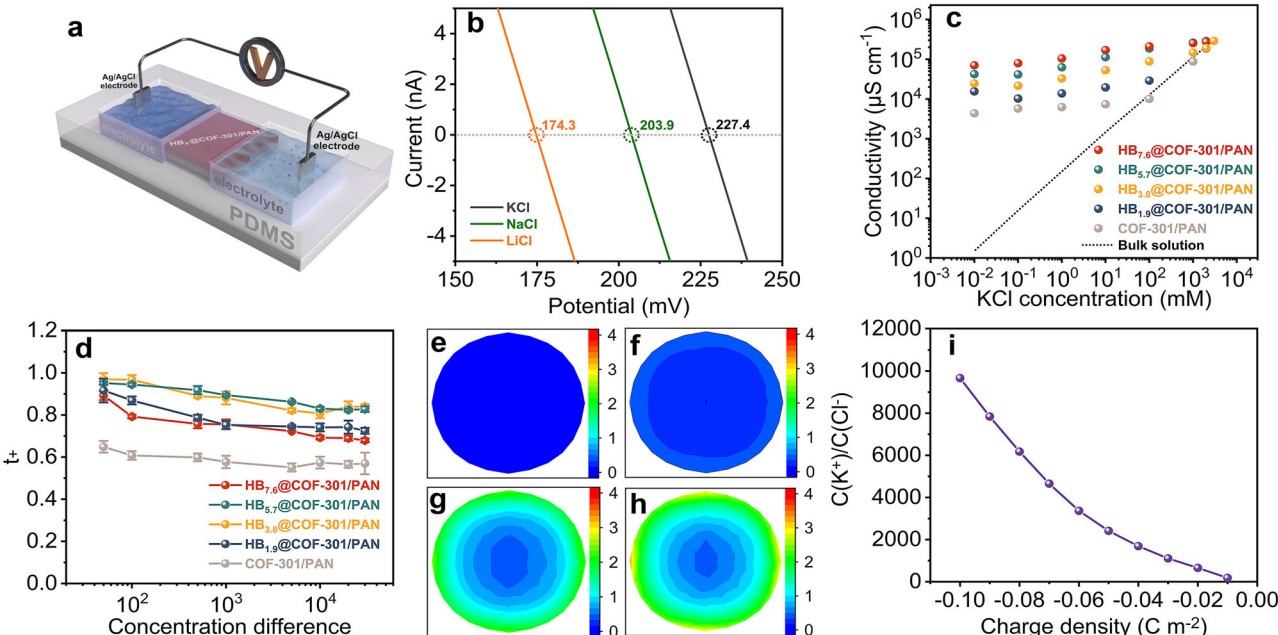

**Fig. 3 | Investigation of the impact of dye content on the transmembrane ion transport. a** Schematic of the experimental setup for measuring the transmembrane ionic transport. **b** $I-V$ plots recorded for the nanofluidic device assembled by HB$_{5.7}$@COF-301/PAN in various electrolytes with a concentration difference of 100. **c** Conductivity versus KCl concentration for the nanofluidic devices assembled by HB$_x$@COF-301/PAN. **d** Plots of transference number ($t_+$) versus various KCl concentration differences across HB$_x$@COF-301/PAN. Error bars depict the standard deviation of three indi-vidual measurements. **e–h** The x–y cross-sectional concentration images of the ratio of K$^+$/Cl$^-$ (the values were divided by 10$^4$) taken at the mouth center facing the low concentration side for COF-301/PAN, HB$_{1.9}$@COF-301/PAN, HB$_{3.8}$@COF-301/PAN, and HB$_{5.7}$@COF-301/PAN, respectively. **i** Plots of the K$^+$/Cl$^-$ ratio at the circle center derived from Supplementary Fig. 33.

pathway, attracting K$^+$ cations while repelling Cl$^-$ anions, thus maintaining nearly constant cationic concentrations. Due to the sub-nanosized pore channels, the concentrations of K$^+$ ions in HB$_x$@COF-301/PAN are significantly higher than those in the bulk, and this discrepancy widens with increasing dye content within the membrane.

To gain a comprehensive understanding of the permselectivity of nanofluidic devices assembled with composite membranes of varying dye contents, we collected the $I-V$ curves under a wide range of KCl concentration differences. One side of the device was filled with 0.1 mM KCl, while the other side featured an increasing concentration from 5 mM to 3 M. Supplementary Fig. 28 presents the plots of open-circuit voltage ($V_{oc}$) and short-circuit current ($I_{sc}$) against the concentration differences for HB$_x$@COF-301/PAN. The calculated cation transfer numbers ($t_+$), summarized in Fig. 3d, unveil that the dye content has a substantial influence on the ion screening capabilities of the nanofluidic device (Supplementary Fig. 29). The pristine COF-301/PAN displayed a slight degree of permselectivity, with $t_+$ values hovering around 0.55 for concentration differences surpassing 10$^3$. This observed permselectivity can be attributed to the slightly negatively charged pore surface resulting from the densely distributed phenolic hydroxyl groups. The inclusion of HB significantly enhanced the permselectivity of the membranes. More specifically, HB$_{1.9}$@COF-301/PAN, featuring an HB content of 0.181 mmol g$^{-1}$, provided a $t_+$ value of 0.74, even at a concentration ratio of 10$^4$. As the dye content increased from 0.181 to 0.324 mmol g$^{-1}$, there was a gradual improvement in permselectivity. However, when the initial dye concentration was further increased to 7.6 mg mL$^{-1}$, a sharp decrease in permselectivity was observed. We attribute this phenomenon to the fact that an excessive amount of HB molecules interferes with the condensation reaction between the –NH$_2$ and –CHO groups, primarily due to the electrostatic interaction between HB and protonated TAM, which disrupts the growth of the membrane. Supporting this explanation, we

observed an increase in absorbance corresponding to the aldehyde C = O group (1735 cm$^{-1}$) of the starting materials as the HB content increased (Supplementary Fig. 30).

To gain a molecular-level understanding of how the dye content influences ion distribution, we conducted numerical simulations using the steady-state Poisson–Nernst–Plank (PNP) equations within the COMSOL Multiphysics software[53]. For these simulations, we designed a cylindrical channel with a length of 100 nm and a diameter of 0.76 nm (matching the COF pore size), which was connected to cylindrical reservoirs on each side, each with a volume of 5 × 10$^4$ nm$^3$ (Supplementary Fig. 31). Supplementary Figs. 32 and 33 present the ion distribution profiles in response to variations in KCl concentration and pore charge density. Given the anionic nature of the dye@COF pore channels, the concentration of K$^+$ ions was notably higher than that of Cl$^-$ ions, and this difference increased as the dye content within the system increased. Furthermore, the plotted relationship between the K$^+$/Cl$^-$ ratio at the channel exit and surface charge density revealed a positive correlation (Fig. 3e–i). These results highlight that increasing the charge density within the nanochannels effectively enhances charge screening and aligns with the trends observed in experimental data.

The exceptional ion selectivity of the resulting nanofluidic devices in high salinity conditions enabled us to harness osmotic energy, creating a self-powered system for microelectronic devices. This is significant because a substantial amount of energy can be harnessed from the entropy change resulting from the mixing of saltwater with varying salinity gradients[54–67]. To assess the efficiency of these newly developed nanofluidic devices and make comparisons with established systems, we conducted tests by applying NaCl solutions at different concentrations (0.5 M NaCl versus 0.01 M NaCl) to mimic the salinity gradients found in estuaries. We determined the maximum output power densities ($P_{max}$) using the formula $P = I^2 R_L$, where $I$ and $R_L$ represent the current density and

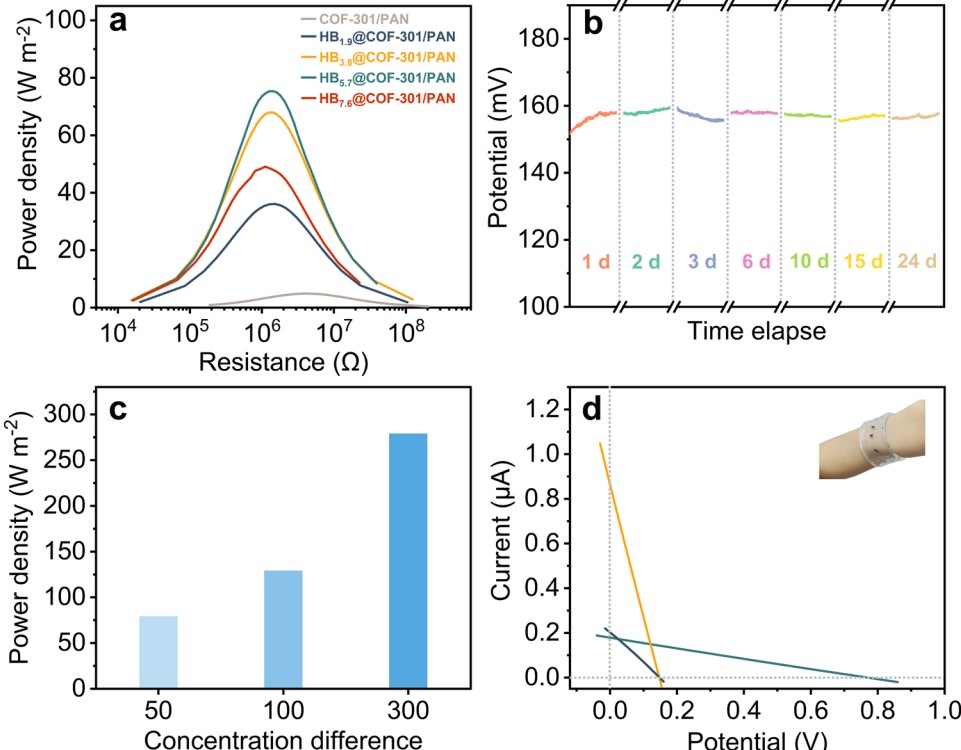

**Fig. 4 | Evaluation of osmotic conversion performance. a** Generated power outputted to an external circuit using $R_L$ with a NaCl concentration difference of 0.01/0.5 M over the nanofluidic devices assembled by HB$_x$@COF-301/PAN. **b** Time-series plots of voltage versus time for the nanofluidic device assembled by HB$_{5.7}$@COF-301/PAN under a NaCl concentration difference of 0.01/0.5 M (testing duration time 30 min). **c** Maximum extractable energy obtained by mixing NaCl solutions with various concentration differences, where the NaCl concentrations in the low concentration side was maintained at 0.01 M, while the other side was increased from 0.5 M to 3 M. **d** The $I-V$ curves of individual devices assembled by HB$_{5.7}$@COF-301/PAN (black line) and its series (olive line) and parallel connections (orange line), where 0.01 and 0.5 M NaCl aqueous solutions were alternatively filled (inset: the patterned nanofluidic device).

oading resistance (the testing area of the membrane was calculated by multiplying its width of 1.5 mm with a thickness of 100 nm, resulting in a value of $1.5 \times 10^{-10}$ m$^2$). The calculated $P_{max}$ values for the nanofluidic devices constructed from COF-301/PAN, HB$_{1.9}$@COF-301/PAN, HB$_{3.8}$@COF-301/PAN, HB$_{5.7}$@COF-301/PAN, and HB$_{7.6}$@COF-301/PAN are 4.9, 36.1, 68.0, 75.4, and 49.1 W m$^{-2}$, respectively (Fig. 4a and Supplementary Fig. 34).

Due to its exceptional charge selectivity and high output power density, HB$_{5.7}$@COF-301/PAN was chosen for subsequent experiments. As depicted in Fig. 4b, the powering process exhibited remarkable stability, with virtually no decay in $V_{oc}$ over a 24-day period. As the salinity gradient increased, the output power density also rose, due to a more substantial driving force. To illustrate, when there was a 300-fold difference in salinity, HB$_{5.7}$@COF-301/PAN achieved an impressive output power density of 279 W m$^{-2}$ (Fig. 4c). Furthermore, leveraging the flexibility and foldability of PDMS, we could customize the circuit of the resulting nanofluidic device, such as creating a circular shape (Fig. 4d, inset). By employing parallel or series connections, the current and voltage of the device can be readily scaled up. Specifically, connecting five HB$_{5.7}$@COF-301/PAN-based nanofluidic devices in parallel and in series allows for amplification of the current density to 0.88 A cm$^{-2}$ and the voltage to 0.77 V (Fig. 4d).

Given the eco-friendly nature of solar energy, research has been keenly focused on materials that facilitate its transformation into electrical power. Subsequently, we investigated how light affects the trans-membrane ion transport in nanofluidic devices. We examined the photoelectric characteristics of the resulting device by recording current–time ($I-t$) and voltage–time ($V-t$) profiles under both light-on and light-off conditions. To simulate natural sunlight, Xenon lamps were employed. The signal changes were recorded using a pair of

lightproof shield-protected Ag/AgCl electrodes submerged in identical KCl aqueous solutions (Supplementary Fig. 35).

The $V-t$ traces revealed that the photoelectric response of HB$_{5.7}$@COF-301/PAN was not instantaneous and exhibited a distinctive ascent and descent regime (Supplementary Fig. 36). Following this initial ascent, the $V-t$ trace remained stable during irradiation and demonstrated a significant reliance on the magnitude of the generated photocurrent and photovoltage concerning the KCl concentration. To be specific, the photocurrent density and photovoltage increased by 40 A m$^{-2}$ and 12 mV, respectively, in the presence of 1 mM KCl, and they surged to 340 A m$^{-2}$ and 45 mV, respectively, when 1 M KCl was used as the electrolyte (Fig. 5a). Furthermore, we observed a direct correlation between the intensity of light power density and the resulting induced photocurrent and photovoltage (Supplementary Fig. 37). Notably, the location of irradiation had minimal influence on the photocurrent and voltage in our system. Instead, the primary factor impacting these parameters was the size of the illuminated surface: a larger area of irradiation led to higher values of photocurrent and voltage (Supplementary Fig. 38). This observation departs from the typical trend observed in many documented light-powered ion-transport systems, where an expansion in the illuminated area often results in a reduction of the photoinduced ionic current[12].

A series of control experiments were conducted to elucidate the mechanism behind the observed opto-ionic effects. It's well-documented in the literature that variations in temperature can influence trans-membrane ion transport, leading to significant changes in conductance[68]. However, in our study, we noticed only a marginal offset in conductance when we raised the temperature of the device by approximately 6 K (it's worth noting that the estimated maximum temperature increase during the testing of both the electrolyte

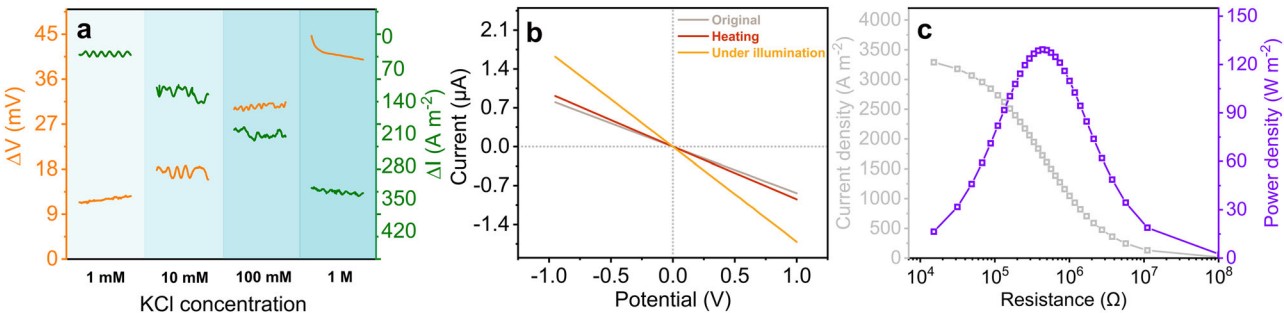

**Fig. 5 | Light-controlled ionic responses and the synchronous harvesting of solar energy and osmotic power.** Ionic responses of the nanofluidic device assembled by HB$_{5.7}$@COF-301/PAN under illumination with 120 mW cm$^{-2}$ of a xenon lamp. **a** Selective region of the increments of voltage and current density measured at different equimolar concentrations of KCl aqueous solutions spanning a range of 1 mM to 1 M (duration time 200 s). **b** $I$–$V$ plots recorded at symmetrical 1 M KCl aqueous solution under various conditions. **c** Output power density with 0.01 and 0.5 M NaCl aqueous solutions.

solution and the membrane was 5.2 K, as indicated in Supplementary Fig. 39). This observation implies that the photogating phenomena we observed cannot be explained by the heat-induced ion transport. In contrast, a clear increase in conductance was observed in the presence of light, indicating an augmentation of surface charge upon light exposure (Fig. 5b)[16–19]. Moreover, when subjecting the nanofluidic device assembled with COF-301/PAN to xenon-light illumination, there was a negligible voltage or current response (Supplementary Fig. 40). These negative control experiments underscore the essential role of HB in explaining the observed photoelectric effect. To further investigate whether the mechanism responsible for the observed photoelectric effect is related to structural changes in HB molecules before and after illumination, we examined UV−vis spectra. After 5 min of irradiation, the intensity of the peaks at 586 and 646 nm decreased rapidly, while the peak at 529 nm increased simultaneously, with a photostationary state reached after 30 min (Supplementary Fig. 41). A similar trend was also observed in the UV−vis spectra of HB$_{5.7}$@COF-301/PAN before and after illumination (Supplementary Fig. 42). For a closer examination of the chemical structure of HB before and after light exposure, we conducted [1]H-NMR spectroscopy. The results showed that the number and relative chemical shift of the hydrogen atoms remained unaltered, indicating the preservation of the chemical structure (Supplementary Fig. 43). However, the chemical shifts of the OH resonances displayed noticeable changes. One of the OH resonances shifted to a higher field, while the other moved to a lower field, suggesting alterations in their spatial positions. DFT calculations provided further insights, revealing significant changes in the lengths of the hydrogen bonds between the phenolic hydroxyl groups and the azo moiety after excitation (Supplementary Fig. 44). These findings collectively suggest that the encapsulated HB molecules undergo proton-coupled electron transfer (PCET) processes upon exposure to light. PCET is a fundamental mechanism observed in bioenergetics and involves the transfer of an electron accompanied by a proton transfer.[69] This process results in the formation of a charge-separated state, facilitating the establishment of a transmembrane potential. Within the host material, COF-301, a well-structured hydrogen-bond network actively promotes proton transfers, thereby facilitating PCET across the membrane and subsequently driving ion transport. Since the photo-driven ion transport is initiated by electron movement, the direction of ion flow aligns with the direction of osmotic ionic flow. Additionally, tests involving alternating light exposure and darkness confirmed the recyclability of the light-triggered ion transport, indicating the structural integrity of the encapsulated dye molecules during the irradiation process (Supplementary Fig. 45).

Recognizing that the generated power can be influenced by both voltage and current, exposure to light augments the surface charge density of the pore channel. This augmentation leads to improved ion selectivity and surface conductance, resulting in increased osmotic voltage and osmotic current. This observation prompted an exploration into concurrently harnessing osmotic power and solar energy. Indeed, there were discernible increments in both $V_{oc}$ and $I_{sc}$ upon light exposure. Specifically, the values of $V_{oc}$ and $I_{sc}$ rose from 147 mV and 0.20 µA to 150 mV and 0.34 µA, respectively. Consequently, the calculated maximum output power density reached 129 W m$^{-2}$, marking a 1.7-fold increase compared to the value achieved without light. Concurrently, the membrane resistance decreased from 1260 kΩ to 435 kΩ. This reduction in resistance illustrates a light-induced efficiency boost for osmotic energy extraction (Fig. 5c).

## Discussion

In summary, this work successfully illustrated a conceptual display of a photo-responsive ionic membrane, achieved through the in-situ host-guest assembly of an angstrom-sized COF host and nano-sized ionic dye molecules. When subjected to light irradiation, the enveloped HB dye molecules engage in PCET processes, thereby initiating the formation of a charge-separated state and enabling the creation of a transmembrane potential. It's worth noting that within the host material, COF-301, a well-organized hydrogen-bond network plays a pivotal role in facilitating proton transfers, which, in turn, enables PCET across the membrane and facilitates ion transport. These feature ensure a direct and positive correlation between the photocurrent and voltage with the irradiation area, irrespective of the irradiation position. As a result, the generated opto-ionic effects significantly enhance ion permeability and reduce membrane resistance, resulting in a substantial increase in the power density of the nanofluidic device. This study not only sheds light on the potential for integrating various energy transduction elements in nanoscale synthetic structures but also offers intriguing and potentially practical applications. Furthermore, considering its generality, the encapsulation of other stimuli-responsive molecules is feasible, promising the membrane additional advanced functionalities for optimized energy extraction.

## Methods
### Materials
Commercially available reagents and solvents were purchased in high purity and used without purification. The asymmetric polyacrylonitrile (PAN) ultrafiltration membrane was obtained from Sepro Membranes Inc. (Carlsbad, CA, USA) with a molecular weight cut off of 50,000 Da.

### Fabrication of HB$_{5.7}$@COF-301/PAN
The PAN support was vertically placed in the middle of a homemade diffusion cell, resulting in each volume of 7 cm$^3$. An acetic acid aqueous solution (1 M, 7 mL) of TAM (21.7 mg, 0.057 mmol) and HB (39.9 mg), and the ethyl acetate and mesitylene solution (V/V = 1/3, 7 mL) of DHA (4.34 mg, 0.026 mmol) were separately introduced into the two sides of the diffusion cell. The reaction mixture was kept at 35 °C for 3 days.

The resulting membrane was rinsed with methanol and ethanol to remove any residual monomers and the catalyst. Finally, each membrane was rinsed with dichloromethane for 24 h and then used for assembling with PDMS elastomer or air-dried for physicochemical characterization.

## Fabrication of nanofluidic devices assembled by HB$_x$@COF-301/PAN

The PDMS was prepared by mixing the prepolymer (SYLGARD™ 184 Silicone Elastomer Base) with the crosslinker (SYLGARD™ 184 Silicone Curing Agent), both from Dow Corning, in a weight ratio of 10:1. The mixture was degassed in an extractor with a vacuum pump and then poured into a poly(methyl methacrylate) testing container. To facilitate handling and experimental procedures, the membrane is typically cut into rectangular shapes measuring 4 mm in length and 1.5 mm in width. Following this initial sizing, the membrane is then encapsulated within the PDMS precursor. The resulting device was cured in oven for 90 min at 75 °C. After cross-linking, two reservoirs of approximately 0.18 cm$^3$ in volume were carved out at the ends of the COF strip. During the process of carving reservoirs within the PDMS, the membrane is further divided into smaller pieces (2 mm in length and 1.5 mm in width). The two ends of the strip were exposed to the various electrolytes and a pair of Ag/AgCl electrodes were inserted into the reservoirs for measurements.

## Characterization of the membranes

X-ray powder diffraction (XRD) patterns were measured with a Rigaku Ultimate VI X-ray diffractometer (40 kV, 40 mA) using CuKα ($\lambda$ = 1.5406 Å) radiation. The gas sorption isotherms were collected on the surface area analyzer ASAP 2020. The N$_2$ sorption isotherms were measured at 77 K using a liquid N$_2$ bath. The dye concentrations were determined by a Thermo Scientific™ UV-Vis spectrophotometers. FT-IR spectra were recorded on a Nicolet Impact 410 FTIR spectrometer. Contact angles of water were measured on a contact angle measuring system SL200KB (USA KNO Industry Co.), equipped with a CCD camera. The static contact angles were measured in sessile drop mode. The CLSM data were collected on a Leica SP5 under an excitation $\lambda_{ex}$ = 405 nm. $^1$H NMR spectra were recorded on a Bruker Avance-400 (400 MHz) spectrometer. Chemical shifts are expressed in ppm downfield from TMS at $\delta$ = 0 ppm, and $J$ values are given in Hz. Scanning electron microscopy (SEM) was performed on a Hitachi SU 8000. XPS spectra were performed on a Thermo ESCALAB 250 with Al Kα irradiation at $\theta$ = 90° for X-ray sources, and the binding energies were calibrated using the C $1s$ peak at 284.9 eV.

## Determination of the content of dye molecules in the membranes

To determine the encapsulated dye content in the resulting membranes, about 10 mg of HB$_x$@COF-301 was digested by 1 M HCl aqueous solution. The concentration of dye in the solution was calculated according to the established standard curve (See details in Supplementary Fig. 12).

## Transmembrane conductance measurement

The ionic current was measured by a CHI660E (CH Instruments). Ag/AgCl electrodes were used to apply a transmembrane potential across the nanofluidic devices assembled by HB$_x$@COF-301/PAN. The membrane was mounted between the two fluidic reservoirs of the conductive cell. Both chambers were filled with symmetric KCl solutions with the concentrations ranging from 0.01 mM to 3 M. To record the $I$−$V$ plots, a scanning triangle voltage signal from −1 V to 1 V with a step voltage of 0.05 V and a period of 1 s was selected. The conductances were derived from the slops of the resulting $I$−$V$ plots. It should be noted that at higher salt concentrations, ion conduction is predominantly governed by bulk conduction. To transforme the

measured conductance values into conductivity, we utilized the bulk conductivity value of a 3 M KCl solution, which was determined separately using a conductivity meter.

## Measurement of the redox potential of the Ag/AgCl electrodes

We utilized both experimental measurements and theoretical calculations to determine the redox potential generated by the unequal potential drop at the electrode-solution interface. For our experimental measurements, we employed a nonselective polyethylene terephthalate (PET) membrane with pores having a diameter of 220 nm. This specific membrane configuration was selected to prevent rapid mixing of the electrolyte solution from the two reservoirs. By using this membrane, we ensured that the measured potential was solely influenced by the asymmetric redox reactions occurring at the electrodes, represented by $\Phi_{redox}$. Remarkably, our experimental results demonstrated that the curve of $\Phi_{redox}$ (also known as $V_{oc}$, open circuit voltage) as a function of concentration differences, with the low concentration side set at 0.1 mM, aligned closely with the values calculated using the Nernst equation (Supplementary Fig. 29). In this manuscript, we utilized the experimental values for all the relevant calculations.

## Charge selectivity evaluation

For investigating the ion transport property of the nanofluidic device assembled by HB$_{5.7}$@COF-301/PAN was contacted with asymmetric electrolyte solutions (100 mM/1 mM, KCl, NaCl, or LiCl). The voltage was scanned with a step of 0.004 V s$^{-1}$ using Ag/AgCl electrodes. The $P_{M^+}/P_{Cl^-}$ selectivity was determined according to the Goldman−Hodgin−Katz equation:

$$\frac{P_{M^+}}{P_{Cl^-}} = \frac{a_{Cl^-,high} \cdot \exp\left(-\frac{V_r F}{RT}\right) - a_{Cl^-,low}}{a_{M^+,high} - a_{M^+,low} \cdot \exp\left(-\frac{V_r F}{RT}\right)} \tag{1}$$

where $a_{Cl^-,high}, a_{M^+,high}, a_{Cl^-,low}, a_{M^+,low}, V_r, F, R, and T$ are the activities of M$^+$ (M$^+$ =K$^+$, Na$^+$, or Li$^+$) and Cl$^-$ in high concentration and low concentration solutions, reverse potential, Faraday constant, gas constant, and temperature, respectively. $V_r$ can be derived from the following equation:

$$V_r = V_{oc} - V_{redox} \tag{2}$$

where $V_{oc}$ is the open-circuit voltage which can be read from the X-intercept of the $I$−$V$ curve, and $V_{redox}$ is the redox potential of Ag/AgCl electrodes under the test concentration gradient.

## Transference number evaluation

For investigating the ion transport property of the nanofluidic devices assembled by HB$_x$@COF-301/PAN, the ion current was recorded by CHI660E. The voltage was scanned with a step of 0.01 V s$^{-1}$ using Ag/AgCl electrodes. The ion transference number $t_+$ of the devices was evaluated by determining the transmembrane diffusion potential ($\Phi_{diff}$). Given that $V_{oc}$ is contributed by the redox potential of Ag/AgCl electrodes ($\Phi_{redox}$) and $\Phi_{diff}$, $t_+$ can be calculated using Eqs. (3) and (4),

$$\Phi_{diff} = V_{oc} - \Phi_{redox} = -\frac{R}{F}\left(T_{low} \ln a_{low} - T_{high} \ln a_{high}\right) \tag{3}$$

$$t_+ = \frac{1}{2}\left(\frac{|\Phi_{diff}|}{\frac{RT}{F} \ln \frac{a_{high}}{a_{low}}}\right) \tag{4}$$

where $V_{oc}$ is the open-circuit voltage, $a_{low}$ and $a_{high}$, and $T_{low}$ and $T_{high}$ are the ion activity and temperature of the high concentration and low

solutions respectively; $R$, $t_+$, and $F$ are gas constant, cation transference number, and Faraday constant, respectively.

## Numerical simulations

The numerical simulation was performed based on coupled Poisson and Nernst−Planck equations by setting appropriate boundary parameters. The Nernst−Planck Eq. (5) defines the flux of each ion species, which describes the transport character of charged nanochannels. The ion concentration induced electrical potential can be described by Poisson Eq. (6). When the system reaches a stationary regime, the ion flux should conform to the steady-state continuity Eq. (7).

$$\vec{J}_i = D_i\left(\nabla c_i + \frac{z_i F c_i}{RT}\nabla\Phi\right) \qquad (5)$$

$$\nabla^2\Phi = -\frac{F}{\varepsilon}\sum z_i c_i \qquad (6)$$

$$\nabla\cdot\vec{J}_i = 0 \qquad (7)$$

where $\vec{J}_i$, $c_i$, $z_i$, and $D_i$ are the ionic flux, ion concentration, charge number of each ionic species $i$, and diffusion coefficient. $\varphi$ and $\varepsilon$ represent the electrical potential and the dielectric constant of medium. $T$, $F$, and $R$ are the absolute temperature, Faraday constant and universal gas constant, respectively. The diffusion coefficients for $K^+$ and $Cl^-$ are $1.9 \times 10^{-5}$ and $2.0 \times 10^{-5}$ cm$^2$ s$^{-1}$, respectively.

For simplification, we use a cylindrical channel of 100 nm in length and 0.76 nm in diameter to simulate the channels inside HB$_x$@COF-301/PAN. To decrease the effect of entrance/exit mass transfer resistances on the overall ionic transport, two electrolyte reservoirs are introduced. The external potential is applied on the boundary $w_1$, and $w_2$ offered the reference potential. The boundary conditions for the electrical potential and ion flux are shown as below:

$$\vec{n}\cdot\nabla\varphi = -\frac{\sigma}{\varepsilon} \qquad (8)$$

$$\vec{n}\cdot\vec{J}_i = 0 \qquad (9)$$

The physical quantity $\sigma$ represents the surface charge density of the channel walls. The ionic current can be calculated by

$$I = F\iint J_i ds = -F\iint D\left(\nabla c_i + z_i c_i \frac{F}{RT}\nabla\varphi\right)ds \qquad (10)$$

## Stability evaluation

Time series of the $V-t$ curves of the nanofludic device were recorded every 24 h after the replacement of new NaCl solutions to ensure that the concentration difference across the membrane was maintained. The initial $V_{oc}$ is 152 mV, which is maintained at 156 mV after 24 d, indicative of the robustness of the nanofludic device.

## Data availability

The data generated in this study are available within the Article, Supplementary Information, or Source Data file. Source data are provided in this paper. The full image dataset is available from the corresponding author upon request. Source data are provided in this paper. Source data are provided with this paper.

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

## Acknowledgements

This work was supported by National Key Research and Development Program of China (2022YFA1503004), National Science Foundation of China (22205198), and National Science Foundation of Zhejiang province (LR23B060001, LY22B06004, and LY23B060022).

## Author contributions

Q.S. and S.W. conceived and designed the research. Q.G. performed the membrane synthesis and most of the testing. Z.L., W.X., S.W., and L.Z. assisted in the membrane characterization. X.Z. carried out the numeric simulation. Z.D. provided valuable suggestions. All authors participated in drafting the paper and gave approval to the final version of the manuscript.

## Competing interests

The authors declare no competing interests.
