## [Peer Review File · Nature Communications]

Photoelectric Responsive Ionic Channel for Sustainable Energy HarvestingREVIEWER COMMENTS

Reviewer #1 (Remarks to the Author):

The manuscript described a new COF membrane structure with dye molecules integrated to form polar channels, presumably by microphase demixing. The system turns out to be permselective for cations and is able to turn osmotic gradients into electric energy, that is to generate "blue energy". When light is shone on the membrane, the current is increasing, thus reacting record high blues energy values which outperform current membranes by a factor of 30.

Of course, I always doubt such high improvement rates, but there is nothing which makes me skeptic in the manuscript so I tend to trust the authors.

For my naivety, I never understood how electric neutrality can be kept by letting only cations pass a membrane, so the voltages must be zero-current polarization voltages, while the current is coming from the depleting reference electrode, and I recommend to make one measurement without reference, as floating potential is no problem in energy generation. But I am sure that my missing education can be compensated by some explanatory sentences also picking up generalist readers like me.

Reviewer #2 (Remarks to the Author):

This manuscript reports a dye-encapsulated covalent-organic framework membrane that shows cation selectivity and can be used in photo-enhanced osmotic energy harvesting. Although a high output power density is achieved in this work, there are some problems with the experimental results and explanations. Therefore, this reviewer recommends a major revision before this manuscript is ready publication. Issues that need to be addressed are listed as follows.

1. In the "Introduction" part (Page 3), it is described that the light absorption will change the redox potential of dye molecules. First, it needs to be further explained here; second, this phenomenon is not mentioned when the experiment results are discussed. Does it have anything to do with photo-enhanced osmotic energy harvesting?
2. The dense layer of PAN substrate is used to load the COF material. However, this dense layer itself is porous and can conduct ions. It seems that this layer will contribute to the ion currents measured in the experiment. In addition, COF can also grow inside this porous layer. More detailed characterization needs to be provided.
3. The thickness of the COF layer in Fig. 2c is difficult to be distinguished.
4. The size of the membrane ("approximately 1.5 and 2.0 mm") is rather small, and it does not seem to match the Fig. S22. How to handle such a small sample in the experiment?
5. The results presented in Fig 3b and Fig. 3d both give the permselectivity of the membrane, but there is a big difference in the results. An explanation needs to be given here. Keep the more reasonable one in the article.
6. For Fig.3c, the details of the calculation of the conductivity should be given.
7. It is described that the membrane structure might be broken with high HB content. Evidence needs to be given here.
8. For osmotic energy harvesting, some important information is missing, including the testing area, the method to obtain the redox potential of the Ag/AgCl electrode, the list of the redox potential under different salt gradient, the method to acquire the osmotic potential and osmotic current. These information is very important and must be described in the article.
9. It is described that the system is stable in 24 days. Considering that the volume of the solution is very small, after 24-day ion diffusion from high-concentration side to the low-

concentration side, will the ion concentration at the of the low-concentration side change? A simple estimate can be provided here.

10. The inset of Fig. 4d seems to have nothing to do with the flexibility and foldability of the device.

11. For photo-induced ion transport, more control experiments need to be provided, such as the influence of wavelength of the light (e.g., absorption wavelengths of dye molecules), the intensity of the light. In addition, if a long-strip membrane is used, how will the current and potential change with light irradiation at different parts.

12. In this work, the light uniformly irradiates on the membrane, so why does it cause ionic flow? What is the direction of the ionic flow? Why is this direction the same as the direction of osmotic ionic flow?

13. The temperature change of the membrane under light irradiation can be characterized using an infrared camera.

14. The HNMR and UV-Vis experiment is done with HB, which is not reasonable. The authors should use HB@COF materials.

Reviewer #1:

Comment 1: The manuscript described a new COF membrane structure with dye molecules integrated to form polar channels, presumably by microphase demixing. The system turns out to be permselective for cations and is able to turn osmotic gradients into electric energy, that is to generate "blue energy". When light is shone on the membrane, the current is increasing, thus reacting record high blues energy values which outperform current membranes by a factor of 30. Of course, I always doubt such high improvement rates, but there is nothing which makes me skeptic in the manuscript so I tend to trust the authors. For my naivety, I never understood how electric neutrality can be kept by letting only cations pass a membrane, so the voltages must be zero-current polarization voltages, while the current is coming from the depleting reference electrode, and I recommend to make one measurement without reference, as floating potential is no problem in energy generation. But I am sure that my missing education can be compensated by some explanatory sentences also picking up generalist readers like me.

Response: We appreciate the reviewer's comments and the support offered for the work conducted in our study. Concerning the issue raised by the reviewer regarding maintaining electric neutrality with only cations passing through the membrane, we have addressed this concern in our study. We integrated a pair of redox electrodes (Ag/AgCl) into the system to facilitate redox reactions that help balance the charge as cations selectively migrate across the membrane and accumulate in the low-salinity reservoir. This configuration ensures that anions remain in the high-salinity reservoir, thereby maintaining charge separation. To provide a clearer understanding of the working principle, we have included a schematic illustration in the revised manuscript. This illustration, found in Supplementary Fig. 1 and cover-letter Figure 1 of both the revised manuscript and the accompanying cover letter, visually demonstrates how the redox electrodes enable the conversion of ionic charge flux into an electrical current while maintaining charge separation and electric neutrality. The redox reactions occurring at the electrodes not only address the concern of electric neutrality but also contribute to the efficient generation of blue energy. By connecting an external load to the system, we can harness the potential energy difference between the high-salinity and low-salinity reservoirs to generate electricity effectively.

Cover-letter Figure 1. (a) Schematic illustration of harvesting energy from salt concentration gradient via the HB_x@COF-301/PAN membrane. A selective membrane favors the transport of counterions from the high-concentration reservoir (left) to the low-concentration reservoir (right) and thus generates an ionic current. The negatively charged HB_x@COF-301/PAN membranes, which favor the flux of cations over anions, are shown. The difference between the cationic flux and anionic flux is the ionic current, I . The Ag/AgCl electrodes are necessary to convert the ionic current to electrical current via redox reactions, thereby closing the electrical circuit. (b) Equivalent circuit representing the harvesting of energy by an external load, R_L , where R_{int} is the internal resistance of the nanopore-based power generator and ΔE_m is the membrane potential.

Reviewer #2:

Comment 1: This manuscript reports a dye-encapsulated covalent-organic framework membrane that shows cation selectivity and can be used in photo-enhanced osmotic energy harvesting. Although a high output power density is achieved in this work, there are some problems with the experimental results and explanations. Therefore, this reviewer recommends a major revision before this manuscript is ready publication. Issues that need to be addressed are listed as follows.

Response: We appreciate the reviewer's high comments and support of our work. The concerns raised by the reviewer have been addressed point-by-point listed as follows.

Comment 2: In the "Introduction" part (Page 3), it is described that the light absorption will change the redox potential of dye molecules. First, it needs to be further explained here; second, this phenomenon is not mentioned when the experiment results are discussed. Does it have anything to do with photo-enhanced osmotic energy harvesting?

Response: We thank the reviewer for these insightful comments. To clarify the concept, we would like to emphasize that dye molecules are typically excited when they absorb light, leading to photo-induced electron transfer. After careful examination of our experimental results, including ¹H NMR, UV-vis spectra, and DFT calculations, we have determined that the encapsulated HB molecules undergo proton-coupled electron transfer (PCET) processes upon light irradiation. PCET involves the movement of an electron from one site to another, accompanied by proton transfer, leading to the generation of a charge-separated state (Acc. Chem. Res. 2018, 51, 445-453). PCET processes are of great significance in living cells as they establish transmembrane gradients of electrochemical potential, serving as a fundamental link between redox processes and bioenergetics. In our study, we propose that these PCET processes generate a built-in electric field across the membrane, which drives ion transport, similar to what is observed in photovoltaic devices. Initially, light is converted into separated charges through PCET, triggering the establishment of a transmembrane potential. This transmembrane potential then facilitates directional ion movement.

Comment 3: The dense layer of PAN substrate is used to load the COF material. However, this dense layer itself is porous and can conduct ions. It seems that this layer will contribute to the ion currents measured in the experiment. In addition, COF can also grow inside this porous layer. More detailed characterization needs to be provided.

Response: We thank the reviewer for the constructive comment. The pores in the PAN membrane are cylindrical in shape, and they are only open for transport in the direction perpendicular to the membrane surface. To avoid any confusion and further elucidate the pore structure, we have revised Fig. 2b, removing the previous schematic diagram of PAN and replacing it with more accurate and descriptive representations of the pore shapes. Additionally, for a comprehensive understanding, we have included a schematic illustration of the pore structure of the PAN in the Supplementary Fig. 14. Furthermore, we have conducted additional experiments using our testing setup to empirically demonstrate that ions cannot be conducted through the PAN membrane (Supplementary Fig. 25). These experiments provided further evidence and validation of our findings regarding the ion transport properties of the membrane.

Comment 4: The thickness of the COF layer in Fig. 2c is difficult to be distinguished.

Response: We appreciate the reviewer's comment. We have replaced the SEM image in Fig. 2c with a new image that allows for easy identification of the thickness of the COF layer.

Comment 5: The size of the membrane ("approximately 1.5 and 2.0 mm") is rather small, and it does not seem to match the Fig. S22. How to handle such a small sample in the experiment?

Response: We thank the reviewer for pointing this out. In order to address any potential confusion, we have made adjustments to accurately reflect the scale of the membrane shown in Supplementary Fig. 24. To facilitate handling and experimental procedures, the membrane is typically cut into rectangular shapes measuring 4 mm in length and 1.5 mm in width. Following this initial sizing, the membrane is then encapsulated within the PDMS material. During the process of carving reservoirs within the PDMS, the membrane is further divided into smaller pieces based on the desired scale for the experiments. We have included these experimental details in the revised manuscript.

Comment 6: The results presented in Fig 3b and Fig. 3d both give the permselectivity of the membrane, but there is a big difference in the results. An explanation needs to be given here. Keep the more reasonable one in the article.

Response: We appreciate the reviewer's comment. The observed discrepancies in the permselectivity of the membrane between Fig. 3b and Fig. 3d can be attributed to the differences in concentration gradients of electrolyte solutions employed in these experiments. We would like to clarify these experimental conditions in our revised manuscript to provide a comprehensive understanding of our findings. In Fig. 3b, we filled two reservoirs with electrolyte solutions of concentrations 0.1 M and 1 mM, respectively. This setup generated a concentration difference of 100 across the membrane. On the other hand, in Fig. 3d, we maintained the concentration of KCl in one reservoir at a fixed concentration of 0.1 mM, while gradually increasing the concentration of KCl in the other reservoir from 5 mM to 3 M. These distinct experimental conditions, characterized by varying concentration differences, account for the observed variations in the membrane's permselectivity between Fig. 3b and Fig. 3d. We have ensured to provide these specific details and explanations in our revised manuscript, allowing readers to better understand the factors influencing the observed permselectivity results.

Comment 7: For Fig.3c, the details of the calculation of the conductivity should be given.

Response: We thank the reviewer for bringing up this point. In our experimental setup, we conducted a study using rectangular strips measuring approximately 1.5 mm in width and 2.0 mm in length embedded within a PDMS elastomer. Two reservoirs were created on either side of the PDMS to expose the two ends of the membrane to KCl aqueous solutions. Each reservoir had a volume of approximately 0.18 cm³. To measure the ionic current, we employed a CHI660E device from CH Instruments. Ag/AgCl electrodes were used to apply a transmembrane potential. In order to obtain the current-voltage (I-V) curves, we applied a scanning triangle voltage signal ranging from -1 V to 1 V, with a step voltage of 0.05 V and a period of 1 second. The conductance values were determined by calculating the slopes of the resulting I-V curves, providing a quantitative measure of ion conduction. We filled both reservoirs with identical KCl aqueous solutions with concentrations varying from 0.01 mM to 3 M. It should be noted that at higher salt concentrations, ion conduction is predominantly governed by bulk conduction. To transform the measured conductance values into conductivity, we utilized the bulk conductivity value of a 3 M KCl solution, which was determined separately using a conductivity meter. We have included these specific details in our revised manuscript to provide a clear description of our experimental procedures.

Comment 8: It is described that the membrane structure might be broken with high HB content. Evidence needs to be given here.

Response: We thank the reviewer for the insightful comment. We conducted IR spectroscopy analysis of HB_x@COF-301/PAN and noticed that the corresponding absorbance of the aldehyde C=O group (1735 cm⁻¹) in HB_x@COF-301/PAN became stronger as the HB content increased. This observation can be rationalized by considering the size of the dye molecule in comparison to the pore size of COF-301. Since the dye molecule is larger than the pore size of the COF-301 framework, it hinders the full condensation of the monomers. As a result, the presence of the dye leads to a higher concentration of unreacted aldehyde groups within the COF

structure, which is reflected in the increased absorbance at 1735 cm^{-1} in the IR spectra. We have incorporated this evidence into our revised manuscript.

Comment 9: For osmotic energy harvesting, some important information is missing, including the testing area, the method to obtain the redox potential of the Ag/AgCl electrode, the list of the redox potential under different salt gradient, the method to acquire the osmotic potential and osmotic current. These information is very important and must be described in the article.

Response: We appreciate the reviewer for bringing up these points. The testing area of the membrane was determined by multiplying its width of 1.5 mm with a thickness of 100 nm, resulting in a calculated value of $1.5 \times 10^{-10}\text{ m}^2$.

We utilized both experimental measurements and theoretical calculations to determine the redox potential generated by the unequal potential drop at the electrode-solution interface. For our experimental measurements, we employed a nonselective polyethylene terephthalate (PET) membrane with pores having a diameter of 220 nm. This specific membrane configuration was selected to prevent rapid mixing of the electrolyte solution from the two reservoirs. By using this membrane, we ensured that the measured potential was solely influenced by the asymmetric redox reactions occurring at the electrodes, represented by Φ_{redox} . Remarkably, our experimental results demonstrated that the curve of Φ_{redox} (also known as V_{oc} , open circuit voltage) as a function of concentration differences, with the low concentration side set at 0.1 mM, aligned closely with the values calculated using the Nernst equation. This observed alignment further validated the accuracy and reliability of our experimental setup and calculations. We have included these experimental details in the revised Supplementary Information. In this manuscript, we utilized the experimental values for all the relevant calculations.

Comment 10: It is described that the system is stable in 24 days. Considering that the volume of the solution is very small, after 24-day ion diffusion from high-concentration side to the low-concentration side, will the ion concentration at the of the low-concentration side change? A simple estimate can be provided here.

Response: We appreciate the valuable comments provided by the reviewer. Our experimental setup involved the use of containers with a small volume (0.18 cm^3) to hold the NaCl aqueous solutions. In order to maintain the stability of the system throughout the 24-day period, we took the necessary precautions to prevent the solutions from drying out. As stated in the Supplementary Information of the original manuscript, we replenished the NaCl aqueous solutions on a daily basis. By regularly replenishing the solutions, we ensured that the volume remained sufficient for the duration of the experiment. Consequently, the concentration of ions at the low-concentration side would not have significantly changed over the 24-day period.

Comment 11: The inset of Fig. 4d seems to have nothing to do with the flexibility and foldability of the device.

Response: We appreciate the reviewer for bringing this to our attention. In response to the comment, we have implemented substantial revisions to our experimental setup. We have developed a device composed of nine interconnected membrane sections arranged linearly. The resulting nanofluidic device can be rolled into a circular shape, highlighting the potential for diverse geometries.

Comment 12: For photo-induced ion transport, more control experiments need to be provided, such as the influence of wavelength of the light (e.g., absorption wavelengths of dye molecules), the intensity of the light. In addition, if a long-strip membrane is used, how will the current and potential change with light irradiation at different parts.

Response: We thank the reviewer for providing valuable suggestions. To emulate sunlight, we employed xenon lamps of different power densities. Our results demonstrated a direct correlation between the light power density and the induced photocurrent and voltage. Notably, the position of irradiation had minimal effect on the photocurrent and voltage in our system. Instead, the primary determinant was the area of the irradiated

surface: a larger irradiation area yielded higher photocurrent and voltage values. These observations have been incorporated into the updated manuscript (Supplementary Figs. 37 and 38, and Cover-letter Figure 2).

Cover-letter Figure 2. Exploring the impact of the illuminated membrane area on the light-induced photocurrent and photovoltage. (a) Selective region showcasing the changes in voltage and current density measured in 10 mM KCl aqueous solutions using the nanofluidic device constructed with HB_{5.7}@COF-301/PAN. The device was illuminated with a xenon lamp at an intensity of 120 mW cm⁻², while the illuminated membrane area varied. (b) Corresponding plots illustrating the average increase in voltage and current density as a function of the ratio of the illuminated membrane area to the total membrane area.

Comment 13: In this work, the light uniformly irradiates on the membrane, so why does it cause ionic flow? What is the direction of the ionic flow? Why is this direction the same as the direction of osmotic ionic flow?

Response: We sincerely appreciate the insightful comments of the reviewer. Our investigations have revealed a significant increase in the photoinduced ionic current as the effective membrane area is enlarged (Supplementary Fig. 38), which diverges from the prevailing trend observed in many documented light-powered ion-transport systems that primarily rely on charge redistribution (e.g., Nat. Commun. 2019, 10, 1171). In contrast, in the second scenario, a decline in the photoinduced ionic current is observed as the area of illumination expands. This decline is attributed to the reallocation of charge carriers, which occurs due to asymmetrical illumination. When this asymmetry in illumination disappears, there is no longer any photoinduced ionic current, consistent with predictions made by the photo-Dember effect.

After carefully examining our experimental results, including ¹H NMR, UV-vis spectra, and DFT calculations, we have determined that the encapsulated HB molecules undergo proton-coupled electron transfer (PCET) processes upon light irradiation. PCET involves the transfer of an electron from one site to another, accompanied by proton transfer, resulting in the generation of a charge-separated state (Acc. Chem. Res. 2018, 51, 445-453). PCET processes play a significant role in living cells as they establish transmembrane gradients

of electrochemical potential, serving as a fundamental link between redox processes and bioenergetics. In our study, we propose that these PCET processes generate a built-in electric field across the membrane, which drives ion transport, similar to what is observed in photovoltaic devices (Angew. Chem. Int. Ed. 2020, 59, 6244-6248). Initially, light is converted into separated charges through PCET, triggering the establishment of a transmembrane potential. This transmembrane potential then facilitates directional ion movement. Furthermore, since the photo-driven ion transport is initiated by electron movement, the direction of ion flow aligns with the direction of osmotic ionic flow.

Comment 14: The temperature change of the membrane under light irradiation can be characterized using an infrared camera.

Response: We appreciate the valuable suggestion from the reviewer. In order to evaluate the temperature change of the HB_{5.7}@COF-301/PAN membrane under 120 mW cm⁻² light irradiation, we employed an infrared camera for monitoring. The results demonstrated a progressive increase in the membrane's temperature during the irradiation process, reaching a steady state after 5 min. We observed a maximum temperature increase of 5.2 K after 5 min of irradiation (see Supplementary Fig. 39).

Comment 15: The HNMR and UV-Vis experiment is done with HB, which is not reasonable. The authors should use HB@COF materials.

Response: We appreciate the comments from the reviewer. In our study, we conducted UV-vis spectra analysis of HB_{5.7}@COF-301/PAN both before and after illumination. After 30 minutes of irradiation, we observed a rapid decrease in the intensity of peaks at 610 and 667 nm, accompanied by an increase in the peak at 490 nm. Although the peak positions of HB in the liquid UV-vis spectra changed to 586, 646, and 529 nm, the overall trend of the changes remained consistent. We attribute these peak position variations to the influence of host-guest interactions between the HB molecules and the COF-301 host, as well as the presence of H₂O in the liquid UV-vis spectroscopy. It is worth noting that different solvents can induce shifts in the absorption maxima or even lead to the formation of distinct species.

Again, we thank the reviewer for the constructive comments and suggestions, which have made our manuscript much improved.

REVIEWERS' COMMENTS

Reviewer #1 (Remarks to the Author):

I went through the modified manuscript and the given responses to the authors, and I am happy with both of them.

This is why I am ready to propose the manuscript for publication as it is. I am sure it will create a lot of discussions within this blue energy community...

Reviewer #2 (Remarks to the Author):

The authors have carefully responded to the raised comments. However, the mechanism of light enhanced ion separation and potential generation needs to be further explained.

It is described that "PCET processes generate a built-in electric field across the membrane". It is worth pointing out that the structure of the membrane device is different from either cell membrane or photovoltaic device. A cell membrane is of a thickness of only several nanometers, and a photovoltaic device consists of asymmetric layered materials. While, the membrane device used in this work is of millimeter scale and has a symmetric structure. The key is to give the reason of why the PCET happened at each small dye molecule can build an electric field inside the millimeter-scale membrane. A further explanation needs to be given at the molecular level.

Reviewer: 1

Comment 1: I went through the modified manuscript and the given responses to the authors, and I am happy with both of them. This is why I am ready to propose the manuscript for publication as it is. I am sure it will create a lot of discussions within this blue energy community.

Response: We appreciate the reviewer's comments and the support offered for the work conducted in our study.

Reviewer: 2

Comment 1: The authors have carefully responded to the raised comments. However, the mechanism of light enhanced ion separation and potential generation needs to be further explained. It is described that "PCET processes generate a built-in electric field across the membrane". It is worth pointing out that the structure of the membrane device is different from either cell membrane or photovoltaic device. A cell membrane is of a thickness of only several nanometers, and a photovoltaic device consists of asymmetric layered materials. While, the membrane device used in this work is of millimeter scale and has a symmetric structure. The key is to give the reason of why the PCET happened at each small dye molecule can build an electric field inside the millimeter-scale membrane. A further explanation needs to be given at the molecular level.

Response: We appreciate the reviewer for their time in reevaluating our manuscript and offering constructive comments. The host material of the membrane, COF-301, can be regarded as an H-bond network, which facilitates proton transfers and consequently enables the occurrence of PCET across the membrane. This observation aligns with our experimental findings, where the intensity of the photoelectric response is solely dependent on the illuminated area and remains unaffected by the specific location of the membrane.

We sincerely thank the reviewer for dedicating time to assess our manuscript and for their valuable constructive comments.